# A unified drug–target interaction prediction framework based on knowledge graph and recommendation system

Qing Ye [1,2,3,6], Chang-Yu Hsieh[4,6], Ziyi Yang[4], Yu Kang [1], Jiming Chen [2], Dongsheng Cao [5✉], Shibo He [2✉] & Tingjun Hou [1,3✉]

Prediction of drug-target interactions (DTI) plays a vital role in drug development in various areas, such as virtual screening, drug repurposing and identification of potential drug side effects. Despite extensive efforts have been invested in perfecting DTI prediction, existing methods still suffer from the high sparsity of DTI datasets and the cold start problem. Here, we develop KGE_NFM, a unified framework for DTI prediction by combining knowledge graph (KG) and recommendation system. This framework firstly learns a low-dimensional representation for various entities in the KG, and then integrates the multimodal information via neural factorization machine (NFM). KGE_NFM is evaluated under three realistic scenarios, and achieves accurate and robust predictions on four benchmark datasets, especially in the scenario of the cold start for proteins. Our results indicate that KGE_NFM provides valuable insight to integrate KG and recommendation system-based techniques into a unified framework for novel DTI discovery.

[1] Innovation Institute for Artificial Intelligence in Medicine of Zhejiang University, College of Pharmaceutical Sciences, Zhejiang University, Hangzhou 310058 Zhejiang, China. [2] College of Control Science and Engineering, Zhejiang University, Hangzhou 310027 Zhejiang, China. [3] State Key Lab of CAD&CG, Zhejiang University, Hangzhou, Zhejiang 310058, China. [4] Tencent Quantum Laboratory, Shenzhen 518057 Guangdong, China. [5] Xiangya School of Pharmaceutical Sciences, Central South University, Changsha 410013 Hunan, China. [6] These authors contributed equally: Qing Ye, Chang-Yu Hsieh. ✉email: oriental-cds@163.com; s18he@zju.edu.cn; tingjunhou@zju.edu.cn

dentification of drug–target interactions (DTI) plays a vital role in various applications of drug development, such as lead discovery, drug repurposing, and elucidation of possible off-target or side effects[1–5]. However, traditional biological experiments for DTI detection are normally costly and time-consuming[6,7]. In the past decades, many computational approaches for DTI identification have been developed to narrow down the search space of drug and protein candidates for reducing cost and accelerating efficiency of drug discovery and development[8–10]. Generally, the approaches for in silico DTI prediction can be classified into three categories: structure-based approaches, ligand-based approaches, and hybrid approaches[11]. The structure-based approaches are not applicable when the three-dimensional (3D) structures of target proteins are unknown and the ligand-based approaches have limited predictive power when there are insufficient bioactivity data for the ligands towards specific targets. The hybrid methods are believed to be more promising to overcome the limitations stated above and to cope with more complex systems by utilizing the information on both drugs and proteins with/without structures. Generally, the hybrid methods can be classified into two subcategories: proteo-chemometrics (PCM) and network-based methods. PCM covers a range of computational approaches developed based on the information of drugs and proteins represented by feature vectors and usually formulate DTI prediction to binary classification[12,13]. This type of approaches allows not only to extrapolate the prediction to discover new compounds toward known targets, but also to extrapolate the prediction to detect new targets toward known compounds. Different machine learning (ML) techniques have been introduced to PCM. Firstly, traditional ML methods, such as support vector machine (SVM) and random forest (RF), have been widely used in this area based on molecular finger-prints and protein descriptors derived from protein sequences[13–18]. Recently, several end-to-end methods based on deep learning (DL), such as DeepDTI and GraphDTA, have been developed for large-scale DTI predictions[19–21].

In addition, network-based methods have been developed by incorporating multiple data sources, such as drug–target inter-actions, drug–drug interactions, and protein–protein interactions, into one framework for DTI prediction. In these networks, nodes can be drugs or proteins and edges are the indicators for the interactions or similarities between the connected nodes[22–26]. In this way, omics data (also called heterogeneous data), such as side-effects, drug-disease associations, and genomics data, have been employed to strengthen DTI prediction. For example, DTINet[27] proposed by Luo et al. applied an unsupervised method to learn low-dimensional feature representations of drugs and target proteins from heterogenous data and predicted DTI using inductive matrix completion. Wan et al. developed an end-to-end method, called NeoDTI, to integrate diverse information from heterogeneous network data and automatically learn topology-preserving representations of drugs and targets to further facil-itate DTI prediction[28]. Thafar et al. combined graph embedding and similarity-based techniques for DTI prediction[29]. Recently, ML models built upon knowledge graph (KG) have been devel-oped rapidly, and quite a few encouraging studies based on KG have been successfully applied to solve many real-world chal-lenges in the development of biomedicine[30–32]. These methods extract the fine-grained multi-modal knowledge elements from omics data and formulate the problem as the link prediction in KG. For example, Mohamed et al. proposed a specific knowledge graph embedding (KGE) model, TriModel, to learn the vector representations for all drugs and proteins and then, consequently, infer new DTI based on the scores computed by the model[33]. For more information about the KG applications in the area of bio-medicine, we refer to the survey article by Zhu et al. that provides a comprehensive review of existing KG-based methods[34]. Another successfully employed technique in DTI prediction is recommendation systems that have become popular and widely applied in various fields, such as e-commerce in the form of web-based software[35,36]. A recommendation system consists of users and objects. Each user collects some objects, for which he/she can also express a degree of preference. The purpose of the algorithm is to infer a user's preferences and provide scores to objects not yet owned, so that the ones, which most likely will appeal to the user, will be rated higher than the others. For the DTI prediction that utilize recommendation systems, the users can be modeled as drugs while the items can be modeled as targets. A mainstream method for recommendations called collaborative filtering has already been integrated with the network-based methods such as dual regularized one-class collaborative filtering[37].

While much effort has been devoted to extracting functional information from heterogeneous data and reducing the noise in heterogeneous networks via matrix decomposition and neural network to further improve prediction performance, there still exists two shortcomings in the above methods: (1) these hybrid methods are highly similarity-dependent and therefore inevitably suffer from activity cliff, which implies that small structural changes can cause large differences in activity[38]. Besides, it is hard to provide a universal definition of similarity for all kinds of omics data collected from various sources, e.g., KEGG Pathway, protein domain and protein binding site. In addition, it is time-consuming to calculate the pairwise similarities for large-scale datasets. (2) Most recent methods are not specifically evaluated in real-world scenarios in which one needs to make DTI prediction when new protein targets are identified for a complicated disease and elucidate molecular mechanisms of drugs with known ther-apeutic effects[39]. This problem, similar to the cold start problem for recommendation systems, is a severe limiting factor for the practical application of DTI prediction methods. As explicated in the subsequent sections, our proposed method performs out-standingly against existing methods in this scenario.

Due to the inevitable noises in the biomedicine data and existing problems stated above, several works such as PharmKG, BioKG, and Hetionet have provided compilations of curated relational data in a unified format, which enables the utilization of multi-omics resources[40–42]. The approaches of utilizing knowl-edge graph could be classified into two types: (1) end-to-end methods based on a comprehensive KG (e.g., DistMult) or a specifically crafted KG focusing on particular downstream tasks (e.g., the work of Zheng et al.[42] designed for drug repurposing and target identification); (2) integration of a pre-trained KGE model and a prediction model toward a specific downstream task. Considering the increasing number and more complex types of biomedical data involved in the knowledge graph, developing a framework that utilizes knowledge graph embeddings in an effi-cient and flexible way is necessary for accurate DTI predictions. Besides, it is also necessary to integrate heterogeneous informa-tion and structural information via multiple approaches and thus enable higher accuracy and broader applications for DTI pre-diction. In this study, we proposed a unified framework called KGE_NFM (Fig. 1) by incorporating KGE and recommendation system techniques for DTI prediction that are applicable to var-ious scenarios of drug discovery, especially when encountering new proteins. KGE_NFM, which could be viewed as a pre-trained model based on knowledge graph and is integrated with a recommendation system tailored for a specific downstream task, captures the latent information from heterogeneous networks using KGE without any similarity matrix and then applies neural factorization machine (NFM) based on recommendation system to enforce the feature representation for a specific downstream task, which is the DTI prediction in this work. The results for the

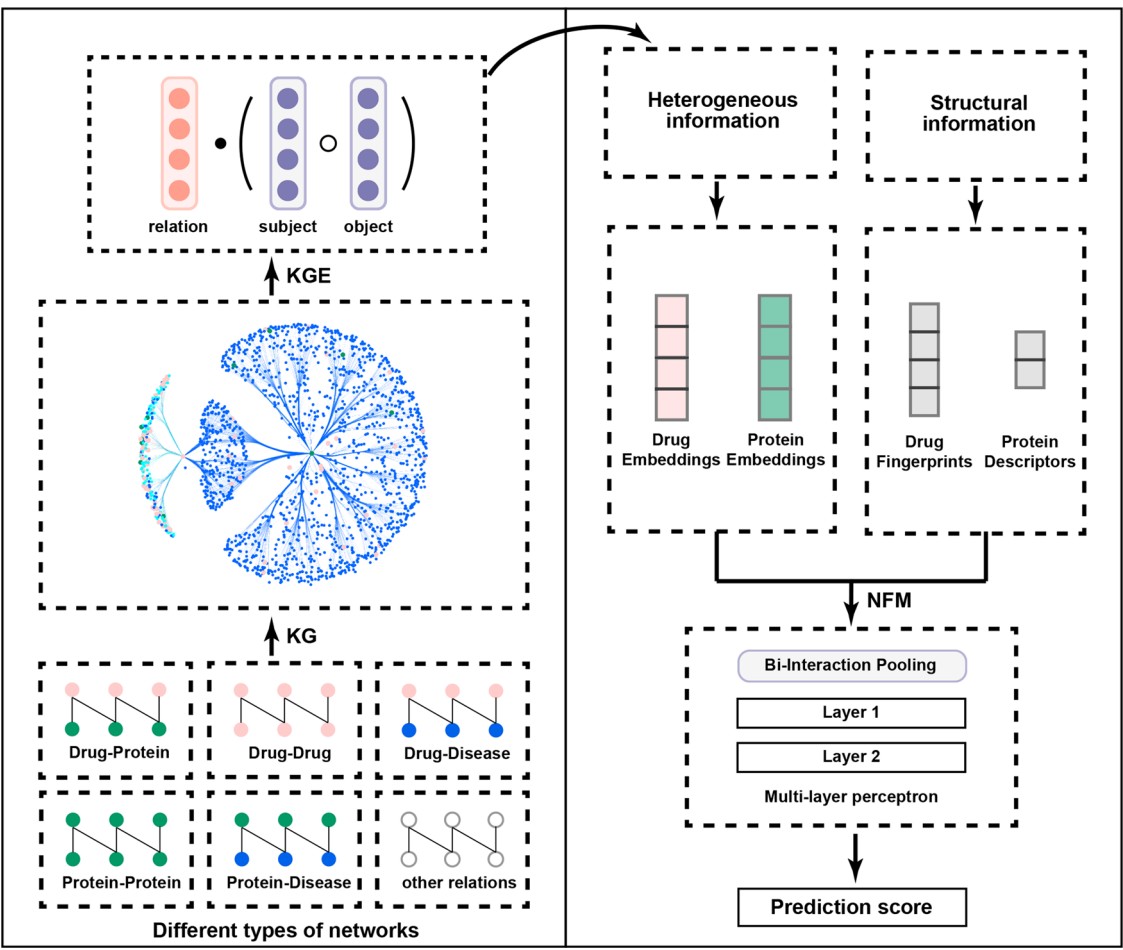

**Fig. 1 The schematic workflow of KGE_NFM.** The pipeline mainly consists of two parts. (1) The construction of KG and embeddings extraction. The original input contains the DTI data and related omics data, and the embeddings of entities and relations are extracted by DistMult. (2) The integration of multimodal information by NFM. The extracted KGEs represent the heterogeneous information, and the molecular fingerprints and protein descriptors represent the structural information. The two types of information are combined and optimized via the Bi-Interaction layer and a feed-forward neural network (FFNN) is used to capture the inherent correlations between DTI.

three common and more realistic evaluation settings toward practical DTI prediction have demonstrated that our method outperformed other baseline methods including feature-based methods, end-to-end ML methods and other network-based methods. Moreover, we have explored the impact of different kinds of KGs on DTI prediction and investigated the effective strategies to make more accurate inferences with KG. All of these results indicate that KGE_NFM is a powerful and robust framework with high extendibility for DTI prediction, which may provide new insights into the novel drug target discovery.

## Results

To evaluate the predictive performance of our method, we compared our method with three types of DTI prediction methods, i.e., feature-based methods, end-to-end methods, and heterogeneous data driven methods. All results were obtained with 10-fold cross-validations. The details of the benchmark datasets (Supplementary Tables 1–4), training procedure, hyper-parameter optimization (Supplementary Table 5) and evaluation results of the four benchmark datasets (Supplementary Tables 6–10) can be found in the Supplementary Materials. KGE and NFM are two main components in our proposed framework, in which KGE is responsible for heterogeneous information integration and NFM is responsible for information extraction that benefits DTI prediction. In the following sections, we present

the performance evaluation on the Yamanishi_08's and BioKG datasets for analyzing the impact of datasets with different size but similar components of KG, and then discuss the approaches that contribute to our extensible framework for the performance improvements of DTI prediction.

**Performance evaluation on the Yamanishi_08's dataset in three sample scenarios.** We compared KGE_NFM with seven baseline methods on the Yamanishi_08's dataset, including MPNN_CNN, DeepDTI, RF, NFM, DTiGEMS+, DistMult and TriModel (Fig. 2, more in Supplementary Table 8).

In the scenario of the warm start, we observed that the heterogeneous data driven methods, DTiGEMS+, TriModel and KGE_NFM, achieved high and robust predictive performance under different ratios between the positive and negative samples (i.e., balanced and unbalanced). Specifically, when the dataset is balanced, the feature-based methods, RF (AUPR = 0.901) and NFM (AUPR = 0.922), and the heterogeneous data driven methods, DTiGEMS + (AUPR = 0.957), TriModel (AUPR = 0.946) and KGE_NFM (AUPR = 0.961), achieve relatively high predictive performance. While for the end-to-end methods, MPNN_CNN (AUPR = 0.788) and DeepDTI (AUPR = 0.820) do not perform as well due to the limited volume of the training set. When the dataset is imbalanced, the AUPR values for the feature-based methods and heterogeneous data driven methods get reduced by different

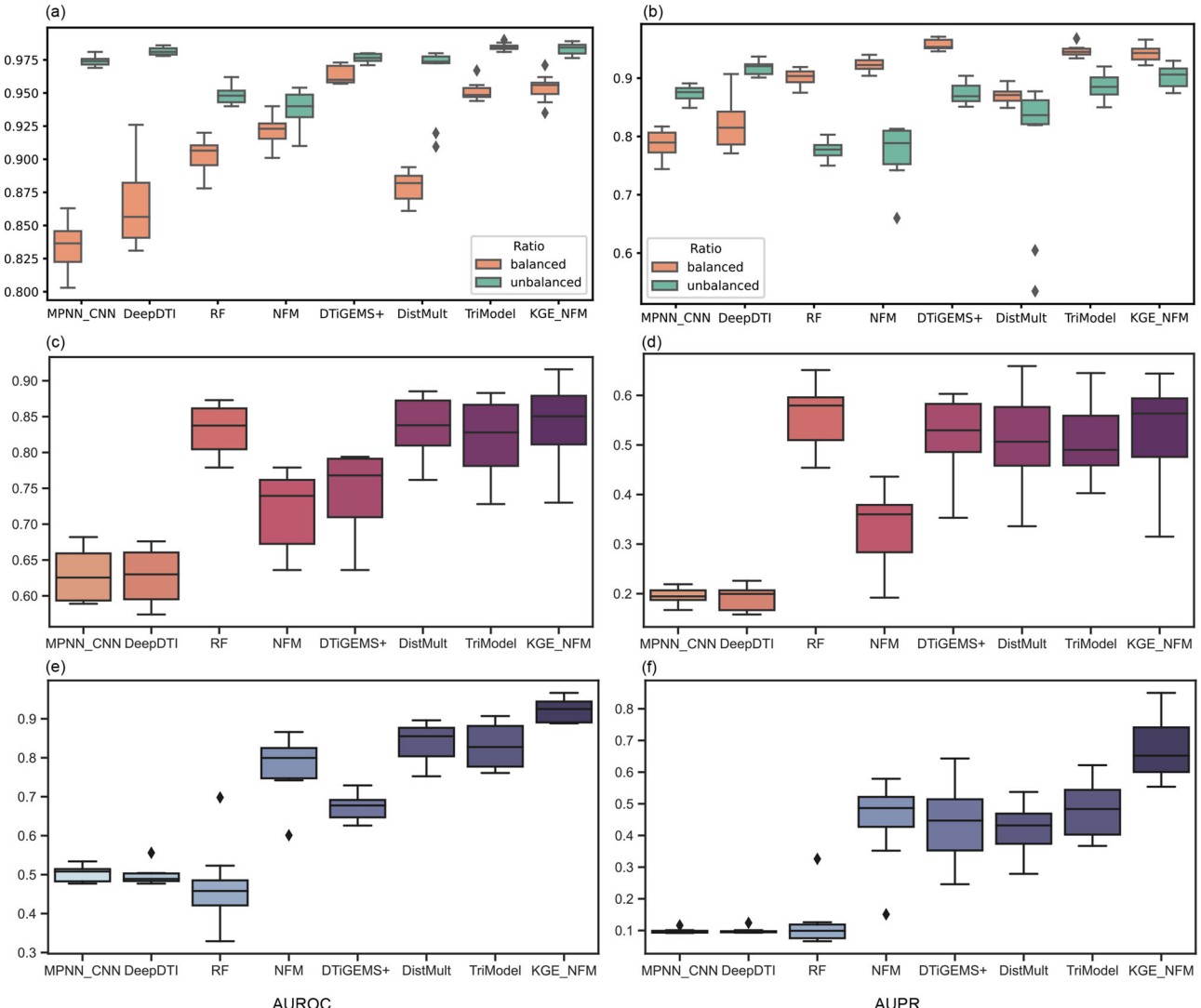

**Fig. 2 Evaluation performance on the Yamanishi_08's dataset in three sample scenarios.** All results were obtained by 10-fold cross-validation. The predictive performance in the scenario of the warm start (Fig. 2a, b) was evaluated with two different ratios between positive and negative samples, in which the 'balanced' means positive:negative≈1:1 and the 'unbalanced' means positive:negative≈1:10. The predictive performance in the scenario of cold start (Fig. 2c–f) was evaluated in the unbalanced situation. $N = 10$ independent experiments. Box plots show the median as the center lines, upper and lower quartiles as box limits, whiskers as maximum and minimum values, and dots represent outliers.

degrees, in which the former decreases over 10% and the later behaves more stably with about 5% decrease. These results indicate that the feature-based methods are prone to be influenced when applying to an unbalanced dataset, while the heterogeneous data driven methods can partly overcome this limitation. As for the end-to-end methods, due to the increased volume of the dataset, their predictive performances increase greatly (about 10% in terms of AUROC and 9% in terms of AUPR) compared to that of the balanced situation. This phenomenon indicates that the end-to-end approaches are limited by the volume of available data; thus, they are more suitable for large-scale DTI predictions.

In the scenario of the cold start for drugs, we observed that KGE_NFM (AUROC = 0.853, AUPR = 0.521) performed best in terms of AUROC, while RF (AUROC = 0.832, AUPR = 0.561) performed the best in terms of AUPR. In comparison between RF and NFM, it seems that the tree-based algorithm is more appropriate than DL models when the structural characterization of drugs (i.e., Morgan Fingerprints) plays the dominant role. In the scenario of the cold start for proteins, KGE_NFM significantly outperformed all the other baselines with a significant leading

margin of 19% in terms of AUPR when compared to the second best performed method TriModel. In comparison between RF and NFM, NFM greatly improves the predictive performance (about 30% in terms of both AUROC and AUPR). This result highlights NFM's potential capability to capture the inherent association in the interactions between drugs and proteins, which provides a huge advantage for NFM in the situation of the cold start for proteins. Then, KGE_NFM, which integrates heterogeneous information with traditional characterization, further improves the predictive performance, 13.5% in terms of AUROC and 21% in terms of AUPR, suggesting that the heterogenous information extracted by KGE is effective for DTI prediction in the scenario of the cold start for proteins. Moreover, it is found that the end-to-end methods did not perform well in the scenarios of the cold start for both drugs and proteins probably due to the extremely different data distributions between the training and test sets. Additionally, we observed similar phenomenon on the four benchmark datasets that KGE_NFM and other heterogeneous data driven methods (DTINet, DTi-GEMS+, DistMult, and TriModel) always performed better in the

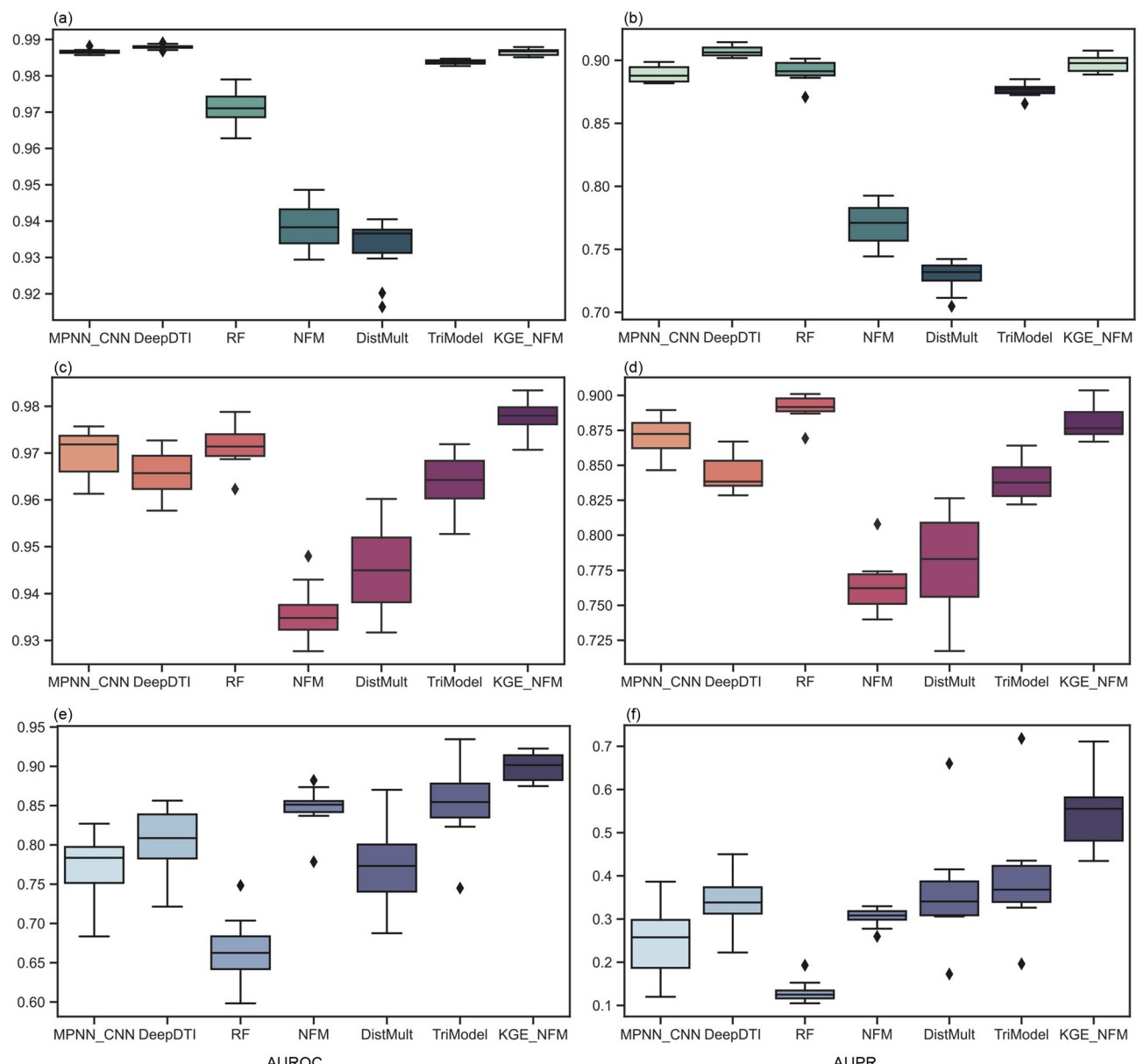

**Fig. 3 Evaluation performance on the BioKG dataset in three sample scenarios.** All the results were obtained by 10-fold cross-validations. The ratio between the positive and negative samples is about 1:10. $N = 10$ independent experiments. Box plots show the median as the center lines, upper and lower quartiles as box limits, whiskers as maximum and minimum values, and dots represent outliers.

scenario of the cold start for proteins rather than the cold start for drugs when comparing with the traditional feature-based method RF. This could probably be attributed to the components of the heterogeneous data, where the protein-related information is more sufficient than drug-related information. For example, there are 83% information is protein-related while only 17% is drug-related in the Yamanishi_08's dataset (Supplementary Table 3). Naturally, KGE will pay more attention on the relationships of proteins in the training process. This finding suggests that the performance of KG-oriented tasks is closely dependent on the components of KG.

**Performance evaluation on the BioKG dataset in three sample scenarios.** We compared KGE_NFM with six baseline methods on the BioKG dataset, including MPNN_CNN, DeepDTI, RF, NFM, DistMult, and TriModel (Fig. 3, more details in Supplementary Table 9).

With a larger size of KG and DTI pairs, the evaluation performance of the baselines under three sample scenarios behaves slightly differently, especially for the end-to-end methods. For the scenario of the warm start, DeepDTI (AUROC = 0.988, AUPR = 0.907) performed the best and KGE_NFM (AUROC = 0.987, AUPR = 0.898) performed the second best. In the scenario of the cold start for drugs, the traditional method RF (AUROC = 0.971, AUPR = 0.891) based on molecular fingerprints and protein descriptors outperformed all the other methods. This phenomenon is also consistent with the other two benchmarks (Tables S6 and S7). This result indicates that it may be enough to use simple feature-based methods like RF in this scenario (more specifically, large-scale virtual screening). In the scenario of the cold start for proteins, KGE_NFM (AUROC = 0.899, AUPR = 0.549) outperformed another heterogeneous data-driven method TriModel with a 15.7% improvement in terms of AUPR. An interesting finding is that the performance of the end-to-end methods greatly improves

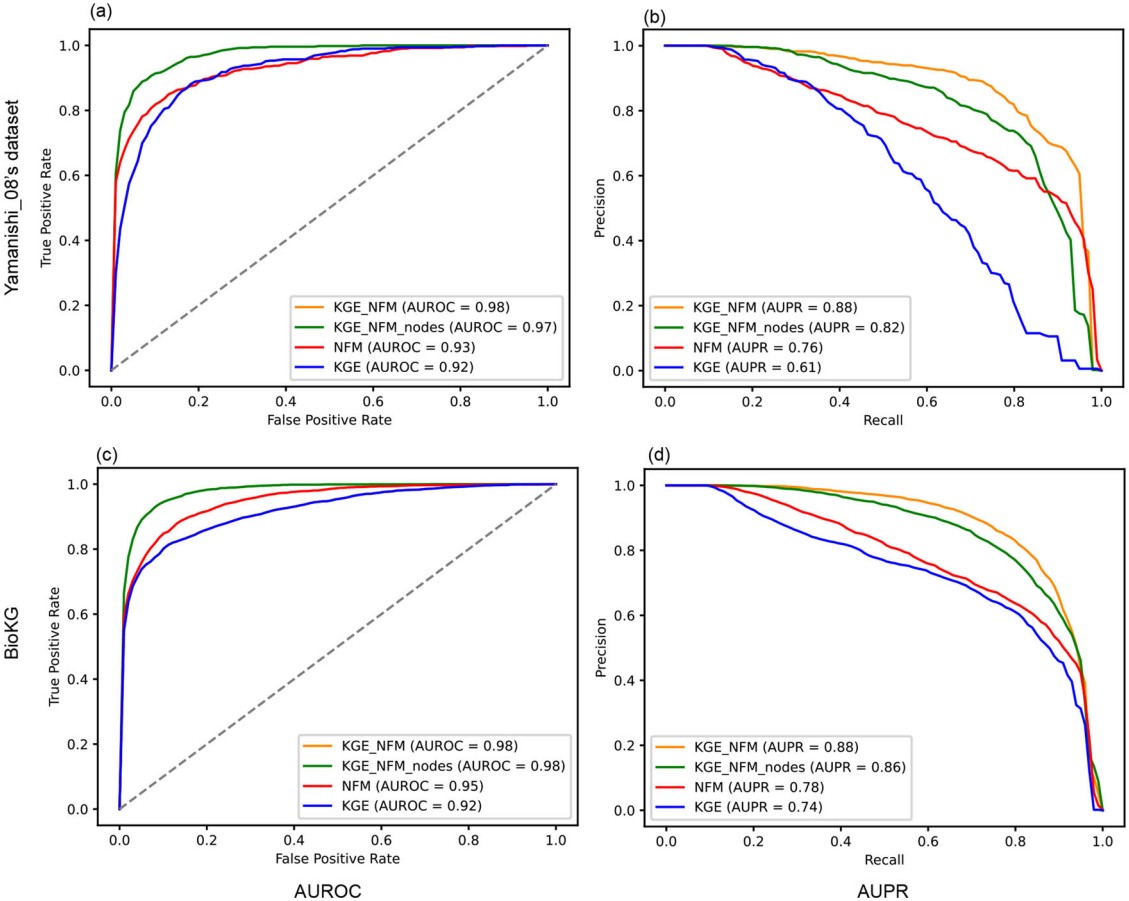

**Fig. 4 Impact of each component in the KGE_NFM framework on predictive performance in the scenario of the warm start in the unbalanced situation.**
**a** and **b** represent the ROC and PR curves on the Yamanishi_08's dataset, respectively. **b** and **d** represent the ROC and PR curves on the BioKG dataset, respectively. Specifically, KGE_NFM_nodes means that the KGE_NFM framework does not incorporate the information of traditional characterization.

in the BioKG dataset compared with the Yamanishi_08's dataset. For example, in the scenario of the cold start for drugs, MPNN_CNN (AUPR = 0.194) did not perform well compared with RF (AUPR = 0.561) in the Yamanishi_08's dataset. While in the BioKG dataset, MPNN_CNN achieved an AUPR of 0.871, which is only 2% less than that of RF (AUPR = 0.891). Similarly, in the scenario of the cold start for proteins, DeepDTI (AUPR = 0.099) performed as poorly as RF (AUPR = 0.117) on the Yamanishi_08's dataset but achieved an AUPR of 0.341 far better than that of RF (AUPR = 0.132) on the BioKG dataset. These findings manifest the influence of the size of datasets on the end-to-end methods, and a large number of drugs and proteins involved in the training set enable the automatically learned features derived from end-to-end methods to behave not badly or even achieve better predictive performance than the handcrafted features (i.e., molecular fingerprints and protein descriptors used in RF) in DTI prediction.

**Impact of each component in the framework on predictive performance.** As Fig. 4 shows that a straightforward application of KGE on DTI prediction (i.e., formulating link prediction problems in a heterogenous graph) does not manifest advantages compared with the feature-based method NFM. In fact, there is a 15% and 4% drop in terms of AUPR on the Yamanishi_08' dataset and BioKG, respectively, when comparing KGE with NFM because of the noises derived from a huge number of heterogeneous information. In this study, we introduced several techniques to overcome this problem and improve the predictive performance. The first one is to apply NFM to infer potential

interactions between drugs and proteins from heterogeneous embeddings. It can be seen from Fig. 4b, d that the predictive performance improves by 21% and 14% in terms of AUPR on the Yamanishi_08' dataset and BioKG, respectively. Besides, we also found that the implementation of traditional characterization of drugs and proteins (KGE_NFM in Fig. 4) also contributes to the predictive performance gain 6% and 2% improvement in terms of AUPR on the Yamanishi_08' dataset and BioKG and makes the prediction more robust (decreased approximately 50% of the standard deviations of both AUROC and AUPR, more details in Supplementary Table 10). These results indicate that our framework is able to efficiently integrate and utilize the information from the structures of biomolecules and omics data for DTI prediction.

**The heterogeneous information extracted from KG contribute to DTI prediction via integrating with other classifiers.** KGE_NFM proposed in this article is an efficient strategy to leverage heterogeneous data for DTI prediction. In fact, KG has tremendous potential for many downstream tasks by incorporating other algorithms in an appropriate way. For instance, we found that the integration of KGE and RF could improve DTI prediction performance compared with RF under three sample scenarios on the Yamanishi_08's dataset. As shown in Fig. 5, both of the AUROC and AUPR of KGE_RF improve compared with those of RF, especially for the scenario of the cold start for proteins, with an increase of 29.2% and 28.2%, respectively.

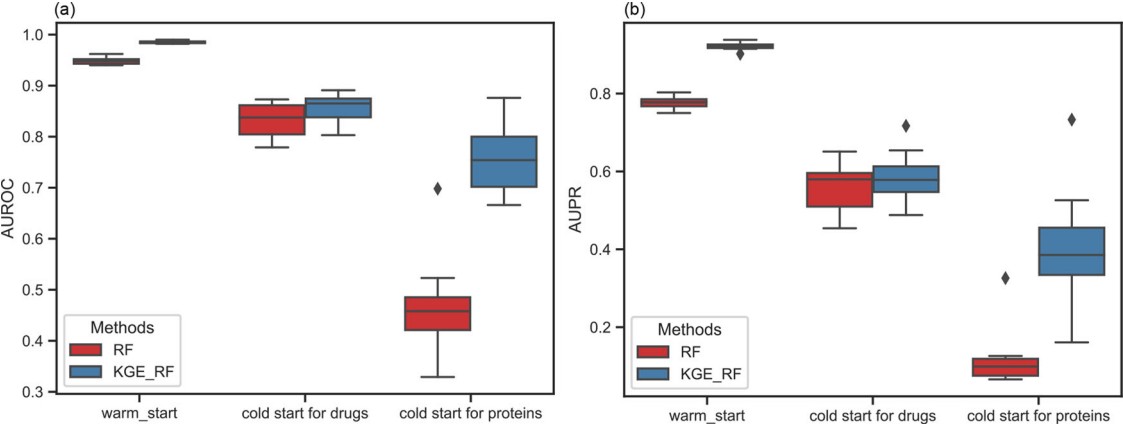

**Fig. 5 KGE enables RF to improve predictive performance on the Yamanishi_08's dataset under three sample scenarios.** KGE_RF uses KGE and drug fingerprints and protein descriptors as the input features and uses RF to build the classifiers. $N = 10$ independent experiments. Box plots show the median as the center lines, upper and lower quartiles as box limits, whiskers as maximum and minimum values, and dots represent outliers.

**Constructing KG in a proper organization could further improve DTI predictive performance**. A systematic integration of biomedical knowledge can enable precise information extraction from heterogeneous data and thus benefit the downstream tasks[41]. Here, to explore how knowledge graph affects DTI prediction, we analyzed the network consisting of DTI data and all other heterogeneous data and harnessed betweenness centrality to measure the centrality of the node in KG (Fig. 6a). Betweenness centrality is equal to the number of shortest paths from all vertices to the others that pass through that node and is often used to identify the nodes that serve as a bridge from one part of a graph to another[43]. Specifically, the betweenness centrality $C_b(n)$ of a node $n$ is computed as follows:

$$C_b(n) = \sum_{s \neq n \neq t} (\sigma_{st}(n)/\sigma_{st})$$

(1)

where $s$ and $t$ are the nodes in the network different from $n$, $\sigma_{st}$ denotes the number of the shortest paths from $s$ to $t$, and $\sigma_{st}(n)$ is the number of the shortest paths from $s$ to $t$ that $n$ lies on.

In the whole network, there are 25,487 unique nodes and most of them own the betweenness centrality values ranging from 0.00–0.02. Only a few nodes have a high value of betweenness centrality, including node identifier (i.e., KEGG_GENE, KEGG_-Drug), KEGG Pathway that represents the knowledge of the molecular interaction, reaction and relation networks (i.e., pathways in cancer), and brite hierarchies (also called KEGG BRITE) that capture the functional hierarchies of various biological objects (i.e., enzymes). These high-centrality nodes that provide generalized type description of related nodes may probably bring useless noises rather than benefits. For example, in the scenario of the cold start for proteins, we chose a test set in 10-fold cross-validation for further exploration and selected one pair of drug-target interaction (D00964–interact–hsa:1553) labeled as a positive in the test set but was predicted as a negative with the prediction probability of 0.14. To figure out the impact of KG on DTI prediction, we picked up the corresponding KG related with the selected DTI. More specifically, we selected the heterogeneous information (called the fist-order KG) related with D00964 and hsa:1553. However, we found that almost no selected node is able to be served as the bridge between D00964 and hsa:1553. Thus, we further selected the heterogeneous information (called the second-order KG) related with the fist-order KG. Then, we analyzed the selected network consisting of the fist-order KG and second-order KG. We observed that the supporting KG did act as a bridge between drugs and proteins but we also found that the selected network seemed to bring in a lot

of noises (Fig. 6b). In the betweenness centrality distribution, we found that the target nodes have low degree of centrality and the betweenness centrality values for D00964 and hsa:1553 are 0.02 and 0.001, respectively. But the nodes like KEGG_GENE and KEGG_Drug, which connect with all genes and drugs, respectively, for node type description, play a dominant role in the selected network and bring in nodes and edges (edges are colored red and shown in the red botted boxes). To overcome this issue, we removed the nodes for identifier including KEGG_GENE, KEGG_Drug and KEGG_PATHWAY, and retrained the KGE_NFM model based on the selected training set. The results show that the prediction performance of the selected DTI pair is improved and the prediction probability reaches 0.95. Similarly, the centrality of the target nodes also improves and the ranking of the betweenness centrality changes from 20 to 8 and 240 to 43 for D00964 and hsa:1553, respectively. Surprisingly, we also found the predictive performance on the whole test set also improved (the value of AUROC holds steady on 0.93 and the value of AUPR changes from 0.69 to 0.73).

## Discussion

In this study, we developed a unified framework, called KGE_NFM, to integrate diverse information from different sources to predict novel DTI. KGE_NFM extracts the heterogeneous information from multi-omics data by KGE and then integrates this information with traditional characterization of drugs and proteins by NFM to yield accurate and robust prediction of DTI. The powerful predictive ability of KGE_NFM has been extensively validated on two benchmark datasets and compared with five state-of-the-art methods under three realistic evaluation settings, especially for the scenario of the cold start for proteins. More importantly, unlike previous methods[27–29], KGE_NFM doesn't rely on similarity networks of drugs and proteins, thus simplifying the integration of multiple types of data. Besides, KGE_NFM can utilize fine-grained heterogeneous information from omics data (e.g., KEGG pathway, protein binding domain). This allows unprecedented applicability of the method to recommend novel DTI within prior knowledge of drugs and proteins. Moreover, we summarized three effective techniques for further improving predictive performance and explained how they impact the prediction in detail. KGE_NFM was shown to be a successful pipeline for DTI prediction by leveraging KG and recommendation system. The analysis demonstrates that NFM, a content-based recommendation system, can efficiently utilize the low-dimensional characterization from KGE and thus significantly improve the prediction

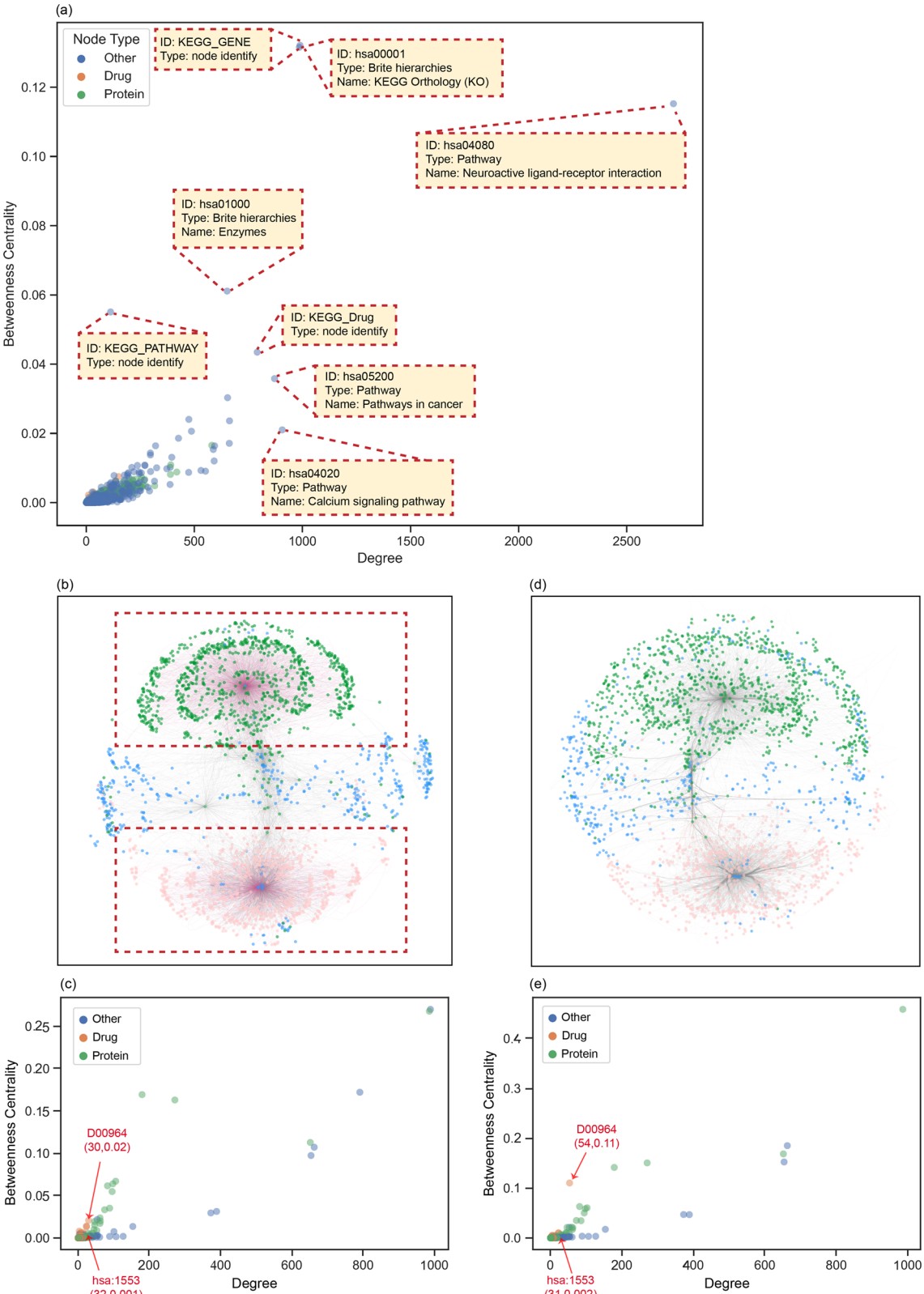

performance. In addition, KGE_NFM is a highly scalable framework and enables the prediction more robust by integrating multi-modal data (i.e., structural information of biomolecules and association information from biochemical networks). Overall, KGE_NFM is a highly competitive approach for DTI prediction

and is promising to facilitate protein target discovery for complicated diseases and molecular mechanisms elucidation, which is a broad and rarely tapped space in computational drug discovery.

While we explain how the removal of noisy nodes contributes to the performance gain in a specific case, this strategy does not

**Fig. 6 Network analyzer and one case to illustrate how to improve DTI predictive performance. a** Betweenness centrality distribution of the network consisting of DTI data and all KG. Degree means the number of the edges linked to a node. The betweenness centrality of a node reflects the amount of the control that this node exerts over the interactions of the other nodes in the network. **b** The visualization of the KG related to the selected DTI (D00964 and has:1553), where the green points represent proteins, the blue points represent heterogeneous information and the red points represent drugs. **c** Betweenness centrality distribution of the network for the KG related to the selected DTI (D00964 and has:1553). **d** The visualization of the selected DTI (D00964 and has:1553) related knowledge graph with removing the nodes and related edges of KEGG_GENE, KEGG_Drug and KEGG_PATHWAY. **e** Betweenness centrality distribution of the network consisting of the selected DTI (D00964 and has:1553) related KG with removing the nodes and related edges of KEGG_GENE, KEGG_Drug and KEGG_PATHWAY.

guarantee substantial gains under all circumstances. As discussed earlier, a systemic organization of biomedical knowledge is crucial for the effective usages of multi-omics data and a more comprehensive investigation in this aspect is planned for a future study. Besides, it should be noted that KGE_NFM is sensitive to the parameter's adjustment and should be treated more carefully during the training procedure. We provided a more exhaustive illustration of the training procedure in the Supplementary Materials. In the future, we will pay more attention to KG construction pipeline in our framework for further improvements of the prediction ability for downstream tasks. We will also expand the application scope of this KG-based recommendation framework in the biomedical science.

## Methods

**Benchmark datasets**. In this study, four benchmark datasets comprising different kinds of heterogeneous data, namely, Luo's dataset, Hetionet, Yamanishi_08's dataset and BioKG, were used to benchmark our method against other state-of-the-art methods for DTI prediction[22,27,40,41].

The Luo's dataset is composed of four types of nodes (i.e., drugs, proteins, diseases, and side-effects) and six types of edges (i.e., drug–target interaction, drug–drug interactions, protein–protein interactions, drug–disease associations, protein–disease associations, and drug–side-effect associations). In total, the network contains 12015 nodes and 1895445 edges (more detailed information in Supplementary Table 1).

Hetionet integrated the biomedical data from 29 publicly available resources and finally obtained 47,031 nodes of 11 types and 2,250,197 relationships of 24 types. Specifically, the nodes consist of 1552 small molecule compounds and 20,945 genes, as well as diseases, anatomies, pathways, biological processes, molecular functions, cellular components, perturbations, pharmacologic classes, drug side effects, and disease symptoms (more detailed information in Supplementary Table 2). It should be noted that the terms "genes" and "proteins" are considered as equal in this study since the proteins are the translation products of genes and most biomedical databases do not distinguish them specifically.

The Yamanishi_08's dataset consists of four sub-datasets: namely, enzymes (E), ion channels (IC), G-protein-coupled receptors (GPCR) and nuclear receptors (NR) collected from various sources including KEGG BRITE, BRENDA, SuperTarget, and DrugBank[44–47]. In this study, we combined the four sub-datasets and the KG was constructed based on the combined dataset. The related heterogeneous data including the ATC codes of drugs, BRITE identifiers, associated diseases and pathways was extracted from KEGG, DrugBank, InterPro, and UniPro by Mohamed et al.[33]. In total, the network contains 25487 nodes and 95579 edges (more detailed information in Supplementary Table 3). The various types of biological information make the biomedical heterogeneous network robust, reusable, and extensible.

BioKG is a biological knowledge graph integrating biomedical data from 14 databases and is designed specifically for relational learning. The contents of BioKG can be categorized into three categories: links, properties, and metadata. Links, e.g., protein–protein interactions and drug–protein interactions, represent the connections between different biological entities. Properties represent the annotations associated to entities and the metadata part contains the data about biological entities, such as names, types, synonyms, etc. As suggested by the original reference[40], not all three parts need to be used for training the KGE model. We only focus on the link part in this study. Thus, KG contains 105524 unique nodes and 2043846 edges (more detailed information in Supplementary Table 4).

**The workflow of KGE_NFM**. KGE_NFM consists of three main components: (1) extraction of heterogeneous information via KGE; (2) automatic dimensional reduction via principal component analysis (PCA); (3) information integration and drug/protein collaborative recommendation via neural factorization machine (NFM).

In the first step, all the related heterogeneous information from different omics (e.g., genomics, proteomics, and metabolomics) were exploited to build a KG, in which each type of biomedical concepts (i.e., drugs, proteins, diseases, and

biological pathways) is considered as a node type and each type of interactions/associations (i.e., drug–protein interactions, drug–drug associations, and protein–pathway associations) is considered as an edge type. The KG stores the information in a triplet form where each triplet represents an interaction/association between two unique entities (e.g., aspirin, drug–target interaction, COX1). After constructing the KG infrastructure, we used a KGE model called DistMult[48] to learn the low-rank representations for all entities and relations. The KGE models generally consist of three steps: (1) the entities and relations are represented in a continuous vector space and initialized as random values; (2) the distance of two entities relative to the relation type is measured via a model-dependent scoring function $f_r(h,t)$ on each triplet $(h,r,t)$, where $h, r, t$ represent head entity, relation, and tail entity, respectively; (3) the output loss is passed to an optimizer in order to update the initial embedding. The goal of the optimization procedure is to assign higher scores to positive samples and lower scores to samples unlikely to be true. DistMult is an extension of RESCAL[49], a semantic matching KGE model that associates each entity with a vector to capture its latent semantics. The score of RESCAL is defined by a bilinear function:

$$f_r(h,t) = \mathbf{h}^T \mathbf{M}_r \mathbf{t} = \sum_{i=0}^{d-1} \sum_{j=0}^{d-1} [\mathbf{M}_r]_{ij} \cdot [\mathbf{h}]_i \cdot [\mathbf{t}]_j \quad (2)$$

where $\mathbf{h}, \mathbf{t} \in \mathbb{R}^d$ ($\mathbb{R}^d$ represents both entities and relations as vectors in the same dimension) are the vector representations of the entities and $\mathbf{M}_r \in \mathbb{R}^{d \times d}$ is a matrix associated with the relation.

DistMult simplifies RESCAL by restricting $\mathbf{M}_r$ to be a diagonal matrix and introduces a vector embedding $\mathbf{r} \in \mathbb{R}^d$ that satisfies $\mathbf{M}_r = \text{diag}(\mathbf{r})$ for each relation $r$. And the score function of DistMult is hence defined as:

$$f_r(h,t) = \mathbf{h}^T \text{diag}(\mathbf{r}) \mathbf{t} = \sum_{i=0}^{d-1} [\mathbf{r}]_i \cdot [\mathbf{h}]_i \cdot [\mathbf{t}]_i \quad (3)$$

This score function captures the pairwise interactions between only the components of $\mathbf{h}$ and $\mathbf{t}$ along the same dimension and thus reduces the computational complexity.

The second step is dimensional reduction through PCA. It is sometimes inappropriate to directly apply the KGE as the input features to the prediction classifier due to the high noise and high dimension of the biological heterogeneous data. To mitigate this potential error, we employ PCA, a popular and effective technique that has been broadly applied in a variety of bio-network related prediction tasks, to process only the relevant entities (e.g., drug and proteins) and retain only the essential aspects of embeddings[50–52]. The introduction of PCA in our framework aims to tune the effective embedding dimension more flexibly and the size of the reduced PCA is considered as a hyper-parameter during the training process of the NFM model.

The third step is to integrate the information from various data sources and make classification via NFM. NFM is a novel extension to factorization machine (FM), which is a popular solution for efficiently using the second-order feature interactions. NFM combines the linearity of FM and the non-linearity of neural network, thus overcoming the issue that FM is insufficient to capture the non-linear and complex inherent structure of real-world data. The scoring function of NFM is:

$$\hat{y}(\mathbf{x}) = w_0 + \sum_{i=1}^{n} w_i x_i + f(\mathbf{x}) \quad (4)$$

where $w_0$ is a global bias, $w_i$ weighs the contribution of the $i$-th feature to the target, $f(\mathbf{x})$ is a multi-layered feed-forward neural network (FFNN) for modeling more complex patterns of feature interactions. Specifically, $f(\mathbf{x})$ contains four parts: (1) embedding layer, a fully connected layer that projects each feature to a dense vector representation,

$$V_x = \{x_1 \mathbf{v}_1, \cdots, x_n \mathbf{v}_n\} \quad (5)$$

where $\mathbf{v}_i$ is the embedding vector for the $i$-th feature and $\mathbf{x}$ is the input feature vector; (2) Bi-Interaction layer, a pooling layer that converts a set of embedding vectors to one vector,

$$f_{BI}(V_x) = \sum_{i=1}^{n} \sum_{j=i+1}^{n} x_i \mathbf{v}_i \odot x_j \mathbf{v}_j \quad (6)$$

where $\odot$ denotes the element-wise product of two vectors, that is,

$(\mathbf{v}_i \odot \mathbf{v}_j)_k = v_{ij}v_{ik}$; (3) hidden layers, a stack of fully connected layers, defined as follows:

$$\mathbf{z}_1 = \sigma_1(\mathbf{W}_1 f_{BI}(V_x) + \mathbf{b}_1)$$
$$\mathbf{z}_2 = \sigma_1(\mathbf{W}_2 \mathbf{z}_1 + \mathbf{b}_2)$$
$$\cdots\cdots$$
$$\mathbf{z}_L = \sigma_L(\mathbf{W}_L \mathbf{z}_{L-1} + \mathbf{b}_L)$$

(7)

where $L$ denotes the number of hidden layers, and $\mathbf{W}_l$, $\mathbf{b}_l$, and $\sigma_l$ denote the weight matrix, bias vector and activation function for the l-th layer, respectively; (4) prediction layer, the output vector of the last hidden layer $\mathbf{z}_L$ which is transformed to the final prediction score:

$$f(\mathbf{x}) = \mathbf{p}^T \mathbf{z}_L$$

(8)

where vector $\mathbf{p}$ denotes the neuron weights of the prediction layer.

**Baselines**. In this work, we evaluated our method against many state-of-the-art methods as the baselines for DTI prediction[19,20,27,29,53,54]. The baselines can be classified into three categories based on their initial input: end-to-end methods use the raw symbols (e.g., SMILES and FASTA sequences) of drugs and proteins as the input, feature-based methods use the molecular fingerprints of drugs and the descriptors of proteins as the input, and heterogeneous data driven methods use the low-dimensional features extracted from heterogeneous data as the input. In this work, we used the Morgan fingerprints calculated by RDKit as the handcrafted featurization for drugs and the CTD descriptors that characterize the compositions, transitions, and distributions of amino acids calculated by PyBioMed as the handcrafted featurization for proteins[55–57].

A summary of the baselines is presented in Table 1. Specifically, the MPNN_CNN and DeepDTI models were constructed with DeepPurpose[53], and the RF model was taken from Scikit-learn[58]. KGE_NFM consists of two parts, in which KGE was constructed with AmpliGraph[59] while NFM was constructed with DeepCTR[60]. More details about the operation and hyperparameter optimization of the baseline methods can be found in Supplementary Table 5.

**Evaluation protocols**. In order to minimize the impact of data variability on the results, 10-fold cross-validation was used to compare the predictive performances of our method and other state-of-the-art methods. Here, we processed the whole knowledge graph into two parts: the task dataset and the supporting knowledge graph. In this work, the task dataset refers to the DTI dataset and the supporting knowledge graph refers to the drug-related information such as drug–drug interactions and protein-related information (e.g., protein–protein interactions). In the training process, (1) the DTI dataset was firstly split into the training set and the test set in each fold according to the scenarios (i.e., warm start, cold start for drugs and cold start for proteins); (2) the supporting knowledge graph and DTIs in the training set were used to train the KGE model; (3) the embedding vectors deprived from the KGE model of the DTIs in the training set and the corresponding descriptors were used to train the NFM model. Then, the model was evaluated on each fold and trained on the other 9 splits. In each training procedure, the known DTI are labeled as the positives while 10 times of the unlabeled DTI were randomly selected to be the negative instances (Supplementary Fig. 1). In this study, we paid a special attention to the differences of the performances for DTI prediction across the following three experimental settings.

Setting I (warm start): Drug repurposing is the most common application for DTI prediction. From the view of safety and development cost, it is a real benefit if the drug that has successfully passed the FDA approval could be used for new diseases[3,61]. Drug repurposing is built upon the hypothesis that drug molecules often interact with multiple protein targets[62]. In this situation, the training and test sets share common drugs and targets.

Setting II (cold start for drugs): For the experimental setting of the cold start for drugs, the test set contains the drugs that are unseen in the training set while all proteins are present in both sets. This scenario is relevant if we need to identify the potential targets that may interact with newly discovered chemical compounds when the 3D structures of targets and the high-quality negative samples are unavailable. For example, GPCRs are the largest super family with more than 800 membrane receptors and over 30% of the approved drugs target human GPCRs[63], but only approximately 30 human GPCRs have solved 3D crystal structures, which limits traditional structure-based drug discovery[64].

Setting III (cold start for proteins): As to the scenario related to the cold start for proteins (discovering new protein targets and elucidation of molecular mechanisms), the test set contains the proteins that are absent in the training set while the drugs are present in both sets. This experimental setting corresponds to a broad application scope, including discovering new protein targets for complicated diseases, elucidating molecular mechanisms of drugs with known therapeutic effects (e.g., active ingredients extracted from Chinese medicine, natural plants or marine organisms), and identifying potential side effects[5,65–68].

It should be noted that the drugs/proteins suffering from cold start problem described in this study only refer to the drugs/proteins existed in the KG but without any known DTI relations. That is to say, we only focus on the cold start problem for drugs/proteins owning available heterogeneous information.

**Table 1 Summary of the baseline methods.**

| Category | Model | Drug featurization | Protein featurization | Heterogeneous information | Classifier |
|---|---|---|---|---|---|
| End-to-end methods | MPNN_CNN | MPNN[19] | CNN | / | MLP |
| | DeepDTI[49] | CNN | CNN | / | MLP |
| Feature-based methods | RF | Morgan fingerprints | CTD descriptors | / | RF |
| | NFM[71] | Morgan fingerprints | CTD descriptors | / | NFM |
| Heterogeneous data driven methods | DTINet[27] (Luo's dataset) | / | / | Network embeddings | Inductive matrix completion[72] |
| | DTiGEMS+[29] (Yamanishi_08's dataset) | / | / | Graph embeddings | MLP |
| | TriModel | Morgan fingerprints | / | KGE | / |
| | KGE_NFM | Morgan fingerprints | CTD descriptors | KGE | NFM |

**Evaluation metrics**. In this study, the performance of each method was evaluated by the area under the receiver operating characteristics curve (AUROC) and the area under the precision-recall curve (AUPR). The receiver operating characteristics (ROC) curve is an efficient indicator for visualizing and measuring the cost of the true positive rate (TPR) against the false positive rate (FPR) at various thresholds[69]. The AUROC of a classifier is equivalent to the probability that a classifier will rank a randomly chosen positive instance higher than a randomly chosen negative instance and is a general measure of the predictive performance for a classifier. Precision-recall curve (PR) shows the tradeoff between precision and recall for different thresholds and a high AUPR represents both high recall and precision[70]. Here, we used AUPR as the main metric for evaluating performance and AUROC as the supplement, since ROC curves are insensitive to the changes in class distribution and the two classes in our study are unbalanced.

**Reporting summary**. Further information on research design is available in the Nature Research Reporting Summary linked to this article.

## Data availability

The source data and data folds in the three sample scenarios used in this study are provided on the Zenodo at https://zenodo.org/record/5500305. The source data of the four benchmarks (the Luo's dataset, Hetionet, the Yamanishi_08's dataset and BioKG) is available on the https://github.com/luoyunan/DTINet, https://het.io/about/, https://drugtargets.insight-centre.org/, https://github.com/dsi-bdi/biokg, respectively. Source data are provided with this paper.

## Code availability

The source data and codes of KGE_NFM are available on the Zenodo at https://zenodo.org/record/5500305.

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

## Acknowledgements

T.H. was financially supported by Natural Science Foundation of China of Zhejiang Province (LZ19H300001), Key R&D Program of Zhejiang Province (2020C03010), and Fundamental Research Funds for the Central Universities (2020QNA7003). S.H. acknowledges support from National Natural Science Foundation of China (62088101).

## Author contributions

T.H., S.H., C.Y.H., and D. C. designed the research study. Q.Y. developed the method and wrote the code. Q.Y., Z.Y., Y.K., and J.C. performed the analysis. Q.Y., S.H., T.H., and C.Y.H. wrote the paper. All authors read and approved the manuscript.

## Competing interests

The authors declare no competing interests.
