## [Peer Review File · Nature Communications]

Reviewers' Comments:

Reviewer #1:

Remarks to the Author:

The paper proposes a new model for drug-target interaction prediction based on combination of knowledge graph embeddings and recommendation system principles. The approach is addressing a very relevant problem, is clearly described and evaluated in comparison with a selection of related, recent tools. The results are competitive and the approach addresses some issues that are not so well covered by existing works (esp. in the context of the "cold start" problem, i.e., predictions of interactions for drug or proteins not seen in the training data). The strengths and weaknesses of the particular approaches are discussed at quite some detail, which may be very useful for people interested in using the models.

Thus the submission may indeed be an interesting contribution to the journal. Before it's ready for publication, however, the issues summarised in the following should be addressed.

MAJOR ISSUES

- While the selection of the baselines for the evaluation seems rather representative of the recent related approaches, I wonder why some of the models reported in Mohamed et al. 2020, or indeed the TriModel itself of those authors, were not included in the study that is otherwise referencing those results multiple times? Some of the models don't have an open source version readily available, but it shouldn't be too complicated to reproduce their results using the published data (similarly to what the authors seem to have done with other models based on the information in the supplement). This would make the validation still more convincing in my opinion.
- The "cold start problem" is only solved in its sort of weaker instance, i.e., for drugs and proteins that occur in the data, only without an explicit link or in different split. Could the method be used also for entirely "out-of-vocabulary" entities? Couldn't the module that produces drug fingerprints and protein descriptors work even without embeddings computed from the network data? If yes, how? If not, why? More details on this practically very relevant topic would be highly appreciated.

MINOR ISSUES

- In lines 222-224, scoring function is said to be a loss function, which is not entirely accurate in my opinion - scoring functions generally combine the head, tail and relation embeddings into a single scalar number, which is then fed into the loss function to maximise the score difference between the positive and negative examples. This should be made clear in the presentation.
- In line 271, I believe the symbol "h" is used for a different vector than before, which is a bit confusing.
- The expression "challenging scenarios" sounds a bit strange - what makes the selected scenarios challenging exactly? Isn't it meant more in the sense of "sample scenarios" or similar?
- Maybe "increases" should be used instead of "increasements" (line 532 for instance), as the latter sounds a bit stilted.
- Figure 2 should come way earlier in the text, close to the part that benefits from that rather key illustration.
- Figure 8 is not too readable - perhaps splitting it and blowing up the parts could help here?

Reviewer #2:

Remarks to the Author:

This paper addresses the issue of Drug-Target Interaction (DTI) by proposing a new method entitled KGE-NFM which combines information learned from a biomedical knowledge graph and

drug/protein features. The KGE-NFM method can be thought of as three major parts. Firstly, a knowledge graph embedding model (the previously proposed DistMult) is trained on a biomedical knowledge graph to learn low-dimensional representations for drug and protein entities. Secondly, these representations are reduced using PCA. Finally, these representations are combined with drug/protein features and passed into a neural factorization machine to finally predict the probability of interaction between a given drug-protein pair. The model is assessed using two benchmark datasets and compared against six methods which encompass different families of approaches (end-to-end, feature based etc). The approaches are compared in different experimental setups where both drugs and proteins are evaluated in the cold-start setup - where entities not seen during the training phase are used in the test. The results show that the proposed approach outperforms others particularly on the cold-start problem for proteins.

The paper is in an interesting area and part of an increasing push for the use of knowledge graphs within drug discovery. I also like the analogies with the recommendation system world. I do have some concerns, which are detailed below, over the method and results which I think would need to be addressed before the paper could be considered for publication. However, I will start with some of the positive aspects of the work:

- The paper is overall written and constructed well and was relatively easy to read and understand both the method and the results.
- Section 3.2.3 has some very interesting results pertaining to the composition of a knowledge graph which I think are very interesting and could even become the main focus of the study.
- The inclusion of a more realistic setup, where proteins and drugs are unseen during the training phase is interesting and under explored in the literature.
- All the model hyper-parameters, as well as the datasets, are fully detailed.

As mentioned, there are some issues with the paper that should be addressed:

- I am concerned with the novelty of the work. The core idea of this paper seems to be in combining graph embeddings with traditional drug/protein features -- however this has been previously explored in prior work. For example, the work of Zheng et al (2020) explored the creation of a biomedical knowledge graph with entity features which is even used in the context of drug repositioning. Yet this approach and others like it are not highlighted in the literature review or compared against. The authors should try and make their novelty more clear given these other works.
- I feel the presented results are rather unconvincing. Indeed, the KGE-NFM only really demonstrates significant increases in performance in the protein cold-start setup with other approaches out performing it in other setups. The authors should try and speculate why this might be and suggest remedies if possible.
- The authors need to further clarify how the datasets were processed. For example, it was not clear where the actual DTI labels were coming from. I assumed these were DTI edges from within the knowledge graphs, however this was not clear. If this is the case, were the DTI edges then removed from the knowledge graph before the DistMult algorithm was used to learn the representations? If not there is significant chance of information bleed from the embeddings. The chosen datasets are also old - why did the authors not also explore the use of other public biomedical knowledge graphs such as Hetionet (Himmelstein et al, 2017) or BioKG (Walsh et al, 2020).
- I feel that some of the choices in the model design are not well motivated enough. Why did the authors feel the need to use PCA to reduce the conditionality of the embeddings? Could a smaller embedding have not been learned by DistMult in the first place? It is also not clear what size the embeddings were reduced to using PCA and the authors did not state if this had an impact on predictive performance. It is also unclear why the authors chose the NFM as the basis of their approach when it is out performed by Random Forest in the results - could the authors add more justification for some of their choices? The authors could also explore better search strategies for the hyper-parameters of their models as a grid search is often too limited to get optimal values. Libraries such as Optuna enable much more principled searches of the hyper-parameter space

(Akiba et al, 2019).

- The baseline approaches I feel could also be improved. For example, why is plain DistMult not directly compared against in Figures 4 & 5?

- Finally I would like to mention reproducibility. The source code is not provided - this method has the potential to be easily replicated, yet without the code, this would be a challenge. However the authors should be commended for including the hyper-parameters they used for training the model.

There are also some minor issues that should be addressed:

- The authors incorrectly name the embedding approach they are using as DisMult when it is really called DistMult (please see the reference (Yang et al, 2015))

- The text on the figures is too small in their current form - please make the figures easier to read.

(Akiba et al, 2019) Akiba, T., Sano, S., Yanase, T., Ohta, T., & Koyama, M. (2019, July). Optuna: A next-generation hyperparameter optimization framework. In Proceedings of the 25th ACM SIGKDD international conference on knowledge discovery & data mining (pp. 2623-2631).

(Himmelstein et al, 2017) Himmelstein, D. S., Lizee, A., Hessler, C., Brueggeman, L., Chen, S. L., Hadley, D., ... & Baranzini, S. E. (2017). Systematic integration of biomedical knowledge prioritizes drugs for repurposing. *Elife*, 6, e26726.

(Walsh et al, 2020) Walsh, B., Mohamed, S. K., & Nováček, V. (2020, October). Biokg: A knowledge graph for relational learning on biological data. In Proceedings of the 29th ACM International Conference on Information & Knowledge Management (pp. 3173-3180).

(Yang et al, 2015) Yang, B., Yih, W. T., He, X., Gao, J., & Deng, L. (2015). International Conference on Learning Representations (ICLR).

(Zheng et al, 2020) Zheng, Shuangjia, et al. "PharmKG: a dedicated knowledge graph benchmark for biomedical data mining." Briefings in Bioinformatics (2020).

The responses to the reviewers' comments

Reviewer #1 (Remarks to the Author):

The paper proposes a new model for drug-target interaction prediction based on combination of knowledge graph embeddings and recommendation system principles. The approach is addressing a very relevant problem, is clearly described and evaluated in comparison with a selection of related, recent tools. The results are competitive and the approach addresses some issues that are not so well covered by existing works (esp. in the context of the "cold start" problem, i.e., predictions of interactions for drug or proteins not seen in the training data). The strengths and weaknesses of the particular approaches are discussed at quite some detail, which may be very useful for people interested in using the models.

Thus the submission may indeed be an interesting contribution to the journal. Before it's ready for publication, however, the issues summarised in the following should be addressed.

Response: We would like to thank the reviewer for a thorough reading of our manuscript and for thoughtful comments and constructive suggestions. We also appreciate that the reviewer agrees this manuscript could be an interesting contribution to Nature Communications. In response to points raised below, we have now supplemented several new benchmarks (along with analyses) and extensively revised our previous manuscript. The detailed point-by-point responses are provided below.

MAJOR ISSUES

(1) While the selection of the baselines for the evaluation seems rather representative of the recent related approaches, I wonder why some of the models reported in Mohamed et al. 2020, or indeed the TriModel itself of those authors, were not included in the study that is otherwise referencing those results multiple times? Some of the models don't have an open source version readily available, but it shouldn't be too complicated to reproduce their results using the published data (similarly to what the authors seem to have done with other models based on the information in the supplement). This would make the validation still more convincing in my opinion.

Response: Thank you for this suggestion and it is very helpful. We have added the evaluation results of TriModel on the Yamanishi_08's dataset in the revised manuscript (Figure 3).

Fig. 3. Evaluation performance on the Yamanishi_08's dataset in three sample scenarios. All results were obtained by 10-fold cross-validation and the bars were expressed as mean \pm SD. The predictive performance in the scenario of the warm start (Fig. 3a and Fig. 3d) was evaluated with two different ratios between positive and negative samples, in which the 'balanced' means positive:negative=1:1 and the 'unbalanced' means positive:negative \approx 1:10. The predictive performance in the scenario of cold start (Fig. 3b, Fig. 3c, Fig. 3e and Fig. 3f) was evaluated in the unbalanced situation.

The results show that KGE-NFM performs better than TriModel on three sample scenarios, especially in the scenario of cold start for proteins (about 19% improvements in terms of AUPR). Besides, we also employed TriModel on the newly added benchmarks (Hetionet and BioKG). The results are consistent with the previous work that KGE-NFM outperforms TriModel in all scenarios, indicating that KGE-NFM is a robust and accurate method for DTI prediction.

(2) The "cold start problem" is only solved in its sort of weaker instance, i.e., for drugs and proteins that occur in the data, only without an explicit link or in different split. Could the method be used also for entirely "out-of-vocabulary" entities? Couldn't the module that produces drug fingerprints and protein descriptors work even without embeddings computed from the network data? If yes, how? If not, why? More details on this practically very relevant topic would be highly appreciated.

Response: We appreciate this valuable comment. One can tackle the challenge raised by the reviewer with our proposed method in various ways. For instance, a feasible solution is to introduce the drug-drug similarity or the protein-protein similarity as additional link types in the knowledge graph. In this way, any new molecule or protein can be integrated with the existing knowledge graph via these similarity links.

In fact, even without introducing the similarity links in the knowledge graph, our method can still work as suggested by the reviewer. Our method utilizes a recommendation algorithm that takes into account of drug fingerprints and protein descriptors, which provides another clue of similarity between entities as defined in a different way. Hence, we can certainly make predictions for these 'out-of-vocabulary' entities even without modifying our method. However, the effectiveness of entirely relying on the fingerprints and descriptors in the recommendation system for cold-start problems are scenario-dependent as detailed in the manuscript. We find it works best in the scenario of drug cold start.

Indeed, several methods, such as DDR [1] and DTiGEMS+ [2], have employed a combination of multiple similarities between drugs and proteins for DTI prediction. Yet, there still remains ample rooms to explore and refine approaches to incorporate similarity-oriented links into a knowledge graph. For example, it has not been systematically assessed which metric for drug/protein similarities work best and how to decide a reasonable threshold for similarity cutoff in order to avoid a knowledge graph with too many similarity edges.

Reference:

[1] Olayan, R. S., Ashoor, H. & Bajic, V. B. DDR: efficient computational method to predict drug–target interactions using graph mining and machine learning approaches. *Bioinformatics* **34**, 1164–1173 (2018).

[2] Thafar, M. A. *et al.* DTiGEMS+: drug–target interaction prediction using graph embedding, graph mining, and similarity-based techniques. *Journal of Cheminformatics* **12**, 1-17 (2020).

MINOR ISSUES

(3) In lines 222-224, scoring function is said to be a loss function, which is not entirely accurate in my opinion - scoring functions generally combines the head, tail and relation embeddings into a single scalar number, which is then fed into the loss function to maximise the score difference between the positive and negative examples. This should be made clear in the presentation.

Response: We agree with the reviewer, and we have refined our wordings accordingly in the revised manuscript.

Action: In Section 2.2 of Pages 8, we revised the loss function description as follows.

“ The KGE models generally consist of three steps: (1) the entities and relations are represented in a continuous vector space and initialized as random values; (2) the distance of two entities relative to this relation type is measured via a model-dependent scoring function $f_r(h, t)$ on each triplet (h, r, t) , where h, r, t represent head entity, relation and tail entity, respectively; (3) the output loss is passed to an optimizer in order to update the initial embedding.”

(4) In line 271, I believe the symbol "h" is used for a different vector than before, which is a bit confusing.

Response: We thank the reviewer for pointing this out. We have corrected the symbol “h” into “p” in the revised manuscript.

Action: In Section 2.2 of Pages 10, we corrected the symbol.

$$f(x) = \mathbf{p}^T \mathbf{z}_L$$

where vector \mathbf{p} denotes the neuron weights of the prediction layer. ”

(5) The expression "challenging scenarios" sounds a bit strange - what makes the selected scenarios challenging exactly? Isn't it meant more in the sense of "sample scenarios" or similar?

Response: Thank you for your comment and it is very helpful. As suggested by the reviewer, we have corrected the “challenging scenarios” into “sample scenarios” in our revised manuscript.

(6) Maybe "increases" should be used instead of "increasements" (line 532 for instance), as the latter sounds a bit stilted.

Response: Thank you for your comment and suggestion. In our revised manuscript, the correction was made.

(7) Figure 2 should come way earlier in the text, close to the part that benefits from that rather key illustration.

Response: Thanks for your suggestion and it is very helpful to improve the paper. We have adjusted Figure 2 to the end of introduction where we firstly propose KGE_NFM and this figure has been updated to Figure 1.

(8) Figure 8 is not too readable - perhaps splitting it and blowing up the parts could help here?

Response: Thank you for pointing this out and it is very helpful. We have reorganized the layout of figures to make it more readable in the revised manuscript.

Reviewer #2 (Remarks to the Author):

This paper address the issue of Drug-Target Interaction (DTI) by proposing a new method entitled KGE-NFM which combines information learned from a biomedical knowledge graph and drug/protein features. The KGE-NFM method can be thought of as three major parts. Firstly, a knowledge graph embedding model (the previously proposed DistMult) is trained on a biomedical knowledge graph to learn low-dimensional representations for drug and protein entities. Secondly, these representation are reduced using PCA. Finally, these representations are combined with drug/protein features and passed into a neural factorization machine to finally predict the probability of interaction between a given drug-protein pair. The model is assessed using two benchmark datasets and compared against six methods which encompass different families of approaches (end-to-end, feature based etc). The approaches are compared in different experimental setups where both drugs and proteins are evaluated in the cold-start setup - where entities not seen during the training phase are used in the test. The results show that the proposed approach outperforms others particularly on the cold-start problem for proteins.

The paper is in an interesting area and part of an increasing push for the use of knowledge graphs

within drug discovery. I also like the analogies with the recommendation system world. I do have some concerns, which are detailed below, over the method and results which I think would need to be addressed before the paper could be considered for publication. However, I will start with some of the positive aspects of the work:

- The paper is overall written and constructed well and was relatively easy to read and understand both the method and the results.
- Section 3.2.3 has some very interesting results pertaining to the composition of a knowledge graph which I think are very interesting and could even become the main focus of the study.
- The inclusion of a more realistic setup, where proteins and drugs are unseen during the training phase is interesting and under explored in the literature.
- All the model hyper-parameters, as well as the datasets, are fully detailed.

Response: Thank you for your positive comments on our work. We believe that we have addressed your concerns in the revised manuscript, and we also provided the detailed point-by-point responses below.

As mentioned, there are some issues with the paper that should be addressed:

(1) I am concerned with the novelty of the work. The core idea of this paper seems to be in combining graph embeddings with traditional drug/protein features -- however this has been previously explored in prior work. For example, the work of Zheng et al (2020) explored the creation of a biomedical knowledge graph with entity features which is even used in the context of drug repositioning. Yet this approach and others like it are not highlighted in the literature review or compared against. The authors should try and make their novelty more clear given these other works.

Response: We thank the reviewer for this useful suggestion. Actually, there are two approaches to utilize knowledge graph (KG) for the biomedical science [1]: (1) end-to-end methods based on a comprehensive KG (e.g., DistMult) or a specifically crafted KG focusing particular downstream tasks (e.g., the work of Zheng et al. [2] designed for drug repurposing and target identification); (2) a combination of a task-independent comprehensive KGE model and a predictive model trained for a specific downstream task (e.g., our proposed KGE-NFM method for DTI belongs to this category). Considering the trend of a fast accumulation of biomedical data and a broad range of complex

problems one can query or make an inference from a KG, we advocate the adoption of the second approach mentioned above.

We believe the advocated approach should be useful in comparison with the first approach. For the first approach, making inference with a task-independent comprehensive KG could be particularly challenging for certain tasks with limited data. Besides, building a task-specific KG (such as PharmKG) requires an expert to decide how to craft a task-specific KG (always some degrees of arbitrariness/bias could be present, i.e., not all experts end up constructing the same task-specific KG) and fine-tune the overall performance with hyper-parameter searching will consume a lot of time with a large KG. Our approach aims to combine the strengths of the other approaches. We use the entire KG to train a task-independent KGE that should faithfully represent the relations between entities. Fine tuning a relatively lightweight model (such as NFM with descriptors for DTI in this case) should be able to achieve a high performance without re-training a potentially large KGE.

Indeed, the evaluation results on the four reported benchmarks (including the updated tests on bigger and more complex KG datasets) have proven the effectiveness of our framework against many top models for DTI predictions. We have supplemented this part in the introduction section of our revised manuscript.

Reference:

[1] Bonner, S. *et al.* A review of biomedical datasets relating to drug discovery: A knowledge graph perspective. *arXiv preprint arXiv:2102.10062* (2021).

[2] Zheng, S. *et al.* PharmKG: a dedicated knowledge graph benchmark for biomedical data mining. *Brief Bioinform* **22**, doi:10.1093/bib/bbaa344 (2021)

(2) I feel the presented results are rather unconvincing. Indeed, the KGE-NFM only really demonstrates significant increases in performance in the protein cold-start setup with other approaches outperforming it in other setups. The authors should try and speculate why this might be and suggest remedies if possible.

Response: We thank the reviewer for this suggestion. The referee implies that KGE-NFM suffers from some inherent flaws, and then we respectfully disagree that KGE-NFM needs a ‘remedy’

although future improvements are certainly necessary just like other published methods. Given the intrinsic complexity of predictive tasks for these networks and KGs, it is difficult to identify one model that could consistently outperform all competing methods across all possible benchmarks and scenarios. As pointed out by the reviewer, KGE-NFM excels in the protein cold-start setup, which is a practical and under-explored scenario for drug discovery. For other drug-discovery scenarios, KGE-NFM also performs competitively against other methods in our study. Hence, we find the word ‘remedy’ too strong as a recommendation in this case.

More formally, we ask the reviewer to check our numerical results in the manuscript again. Please note that KGE-NFM did not consistently outperform by a particular model across all benchmarks. Please note again that whenever some models outperform KGE-NFM, the gap was not significant enough to dismiss KGE-NFM as a second-tier or an unreliable method (with respect to the performance of the top models tested in this study). Hence, given the impressive performance on the protein cold-start setup, we hold onto our view that KGE-NFM is a great DTI predictive model that has a wider range of applicability than most existing models.

Nevertheless, we thank the reviewer for this thoughtful question. We provide a comprehensive analysis. To begin, we share our perspective on how KGE-NFM performs by: (1) comparing with the feature-based methods and end-to-end methods: the protein embeddings derived from the KG are more effective than the protein descriptors and automatically learned vector representations derived from just the protein sequences. In other words, the information from protein functional annotations (derived from the KG) is more useful for DTI predictions rather than the information from 2D protein sequences; (2) comparing with the KGE methods: with the same effective protein embeddings, NFM enables better predictive performance via focusing on the DTI predictions while the KGE methods optimize the objective function on multiple biomedical relations (e.g., Drug-Drug interaction and Protein-Protein interaction). It should be noted that relations in the biomedical KG are normally unbalanced and DTI data only occupy a small part in the whole KG. Based on the above two reasons, KGE-NFM achieves better performance compared with the different kinds of DTI prediction methods for the protein cold-start cases. This observation concurs with our design philosophy (given in our first response) on how KGE-NFM could outperform other methods. This advantage becomes less effective when one tests on well-studied protein targets with many known drug interactions to provide direct clues about DTI. Methods like RF based on molecular and protein

descriptors can perform slightly better than our method in this case. Nevertheless, we argue that KGE-NFM is a more interesting model to be used when one is asked to recommend potential drugs for a newer target.

(3) The authors need to further clarify how the datasets were processed. For example, it was not clear where the actual DTI labels were coming from. I assumed these were DTI edges from within the knowledge graphs, however this was not clear. If this is the case, were the DTI edges then removed from the knowledge graph before the DistMult algorithm was used to learn the representations?

Response: We thank the reviewer to point out the unclarity of the description for dataset processing. Actually, the datasets were processed in the following steps: (1) the DTI data is split into the training set and the test set in each fold according to the scenarios; (2) the supporting knowledge graph and DTIs in the training set are used to train the DistMult model; (3) the DTIs in the training set and the corresponding structural characterization (molecular fingerprints and protein descriptors) are used to train the NFM model; (4) finally, the test set is used to evaluate performance of KGE_NFM. Therefore, there is no information bleed from the embedding in our work since the test set is not involved in the training process.

(4) If not there is significant chance of information bleed from the embeddings. The chosen datasets are also old - why did the authors not also explore the use of other public biomedical knowledge graphs such as Hetionet (Himmelstein et al, 2017) or BioKG (Walsh et al, 2020).

Response: Thank you for this suggestion and it is very helpful to improve the paper. We have added the introduction, statistics information and evaluation results on the dataset “Hetionet” and “BioKG” in the revised manuscript and supplementary materials (Table. S7 & S9).

Table S7. Evaluation performance on the Hetionet in the three sample scenarios.

Metrics	Scenario	End-to-end methods		Feature-based methods		Heterogeneous data driven methods	
		MPNN_CNN	DeepDTI	RF	NFM	TriModel	KGE_NFM
AUROC	Warm start	0.960	0.963	0.958	0.911	0.963	0.972
	Cold start for drugs	0.929	0.891	0.934	0.872	0.908	0.900

	Cold start for proteins	0.480	0.521	0.516	0.582	0.820	0.878
	Warm start	0.721	0.756	0.739	0.610	0.746	0.780
AUPRC	Cold start for drugs	0.641	0.564	0.689	0.563	0.554	0.577
	Cold start for proteins	0.082	0.096	0.086	0.139	0.250	0.408

Table S9. Evaluation performance on the BioKG in the three sample scenarios.

Metrics	Scenario	End-to-end methods		Feature-based methods		Heterogeneous data driven methods	
		MPNN_CNN	DeepDTI	RF	NFM	TriModel	KGE_NFM
	Warm start	0.987	0.988	0.971	0.938	0.984	0.987
AUROC	Cold start for drugs	0.970	0.966	0.971	0.936	0.964	0.978
	Cold start for proteins	0.768	0.806	0.666	0.847	0.852	0.899
	Warm start	0.889	0.907	0.891	0.769	0.876	0.898
AUPRC	Cold start for drugs	0.871	0.844	0.891	0.764	0.839	0.881
	Cold start for proteins	0.245	0.341	0.132	0.303	0.392	0.549

In comparison with five baselines, KGE-NMF performs well under all scenarios on the new benchmarks. Especially, for the scenario of cold start for proteins (~15% improvement compared with the second best). The evaluation results further strengthen our claim that KGE-NMF is a highly accurate and robust approach for DTI prediction.

(5) I feel that some of the choices in the model design are not well motivated enough. Why did the authors feel the need to use PCA to reduce the conditionality of the embeddings? Could a smaller embedding have not been learned by DistMult in the first place? It is also not clear what size the embeddings were reduced to using PCA and the authors did not state if this had an impact on predictive performance. It is also unclear why the authors chose the NFM as the basis of their approach when it is out performed by Random Forest in the results - could the authors add more justification for some of their choices? The authors could also explore better search strategies for the hyper-parameters of their models as a grid search is often too limited to get optimal values. Libraries such as Optuna enable much more principled searches of the hyper-parameter space (Akiba et al, 2019).

Response: We thank the reviewer for pointing out the unclarity with PCA in the KGE-NMF. Hence we more carefully layout the design philosophy of KGE-NFM in the revised manuscript, including

explaining why for PCA and NFM. We summarize the message below.

PCA, introduced in the KGE-NFM framework, aims to tune the effective embedding dimension more flexibly. In other words, the reduced dimension (by PCA) is considered as a hyper-parameter for the inputs to the downstream NFM model. As discussed in our design philosophy, we do not want to re-train the KGE for different tasks. The benefits of utilizing PCA can be verified. Here, we visualize the embeddings for drugs and proteins in the Luo's dataset (Figure. R1). It can be seen that the protein nodes in red boxes are more clearly distinguished from the drug nodes after we apply the PCA.

Figure. R1. Visualization of the drugs (red) and proteins (green) in DTI pairs on the Luo's dataset using the t-SNE for (A) KGE before PCA and (B) KGE after PCA.

While we agree that the performance of KGE with reduced size (re-trained each time for a specific downstream task) can very well match the performance of a higher-dimensional KGE followed by a task-dependent PCA (i.e., processed only the relevant entities for the downstream task), this is not really the focus of our intention. We hypothesize it's better to use a higher-dimension KGE that allows us to more easily train and embed all possible biomedical relations inherent to the KG, and a proper PCA can give us a more focused low-dimensional representation useful for a specific task such as the illustration conveyed in Figure S1.

The adoption of NFM is motivated by an observation: among all kinds of DTI predictive frameworks, methods that takes advantage of the recommender system approaches have outperformed other methods like RF in terms of accuracy for the cold-start scenario [3, 4], which is also consistent with our own results. As cold start is highly relevant for the drug discovery, we are

quite satisfied with the KGE-NFM's performance in all four reported benchmark tests (particularly, the protein cold-start cases) in the revised manuscript.

Finally, we thank the reviewer for suggesting the hyper-parameters optimization strategy and the library Optuna. We have employed it in our newly added two benchmarks (Hetionet and BioKG). With the larger size of DTIs and KG of the new benchmarks, KGE-NFM still performs well and achieves high and robust predictive performance in three sample scenarios, especially in the scenario of cold start for proteins (about 15% improvements in terms of AUPR on the two benchmarks).

Reference:

[3] Bagherian, M. *et al.* Machine learning approaches and databases for prediction of drug–target interaction: a survey paper. *Briefings in bioinformatics* **22**, 247-269 (2021).

[4] Adomavicius, G. & Tuzhilin, A. Toward the next generation of recommender systems: A survey of the state-of-the-art and possible extensions. *IEEE transactions on knowledge and data engineering* **17**, 734-749 (2005).

(6) The baseline approaches I feel could also be improved. For example, why is plain DistMult not directly compared against in Figures 4 & 5?

Response: We accept the reviewer's suggestion and have added the performance of TriModel and DistMult in the revised manuscript (Figures 3 & 4) and supplementary materials (Table S10).

Fig. 3. Evaluation performance on the Yamanishi_08's dataset in three sample scenarios. All results were obtained by 10-fold cross-validation and the bars were expressed as mean \pm SD. The predictive performance in the scenario of the warm start (Fig. 3a and Fig. 3b) was evaluated with two different ratios between positive and negative samples, in which the 'balanced' means positive:negative=1:1 and the 'unbalanced' means positive:negative \approx 1:10. The predictive performance in the scenario of cold start (Fig. 3c, Fig. 3d, Fig. 3e and Fig. 3f) was evaluated in the unbalanced situation.

Fig. 4. Evaluation performance on the BioKG dataset in three sample scenarios. All the results were obtained by 10-fold cross-validations. The ratio between positive and negative samples is about 1:10.

Table S10. Impact of each component in the KGE_NFM framework on the predictive performance in the scenario of the warm start in the unbalanced situation.

Dataset	Metrics	DistMult	NFM	KGE_NFM_nodes	KGE_NFM_nopca	KGE_NFM
Luo's dataset	AUROC	0.814	0.923	0.931	0.913	0.962
	Std	0.023	0.012	0.015	0.022	0.008
	AUPR	0.426	0.743	0.762	0.648	0.855
	Std	0.057	0.045	0.068	0.089	0.035
Hetionet	AUROC	0.959	0.911	0.970	/	0.972
	Std	0.002	0.003	0.002	/	0.001
	AUPR	0.691	0.610	0.774	/	0.780
	Std	0.009	0.009	0.006	/	0.008

	AUROC	0.963	0.939	0.970	/	0.983
Yamanishi_08'	Std	0.026	0.013	0.006	/	0.004
s dataset	AUPR	0.792	0.773	0.826	/	0.902
	Std	0.120	0.048	0.035	/	0.019
	AUROC	0.933	0.938	0.983	/	0.987
BioKG	Std	0.008	0.006	0.002	/	0.001
	AUPR	0.729	0.769	0.888	/	0.898
	Std	0.013	0.017	0.009	/	0.007

As shown in Figure 3 & 4 and Table S10, the proposed KGE-NFM method outperforms TriModel and DistMult in the metrics of AUROC and AUPR on four benchmarks. These results further indicate that KGE-NFM is a reliable method for DTI detection.

(7) Finally I would like to mention reproducibility. The source code is not provided - this method has the potential to be easily replicated, yet without the code, this would be a challenge. However the authors should be commended for including the hyper-parameters they used for training the model.

Response: Thank you for your comment and it is very helpful for us. We agree that sharing the datasets and source codes is important and we have uploaded our original dataset, associated molecular fingerprints, and protein descriptors, and representative scripts used for dataset splitting, hyper-parameters optimization, training and evaluation in this study at <https://zenodo.org/record/5500305>.

There are also some minor issues that should be addressed:

(8) The authors incorrectly name the embedding approach they are using as DisMult when it is really called DistMult (please see the reference (Yang et al, 2015))

Response: We thank the reviewers' for his/her suggestion and we have carefully fixed the typo in the revised manuscript.

(9) The text on the figures is too small in their current form - please make the figures easier to read.

Response: Thank you for your comment and it is very helpful for us. We have adjusted the text in all the figures in the revised manuscript.

Reviewers' Comments:

Reviewer #1:

Remarks to the Author:

Thank you for thoroughly addressing all the important issues raised by the review(s). I have no further comments to add.

Reviewer #2:

Remarks to the Author:

I thank the authors for the revised and improved version of the manuscript. I have a few further thoughts after considering the rebuttal and revised manuscript.

It is still unclear how the data is split and preprocessed. In the rebuttal, the authors state that the protein and drug embeddings are fed into the NFM along with the structural information features. What is still not clear and is crucial for there to be clarity on is are any links within the knowledge graphs between that particular protein and drug removed before the embeddings are removed? This could lead to a problem of trivial predictions where the embeddings between those two entities already encode that they should be linked.

I would still like to see some hypotheses as to why the model does not perform better for the cold start on drugs given that the knowledge graphs should be able to capture information about them.

Lastly, on rereading the article, it seems the authors provide limited justification as to why they chose DistMult over the myriad of other KGE models in the literature. How would the results change in TransE or RotatE was chosen instead?

Our responses to the reviewers' comments are as follows:

Reviewer #1 (Remarks to the Author):

Thank you for thoroughly addressing all the important issues raised by the review(s). I have no further comments to add.

Response: We appreciate that the reviewer agrees this manuscript could be an interesting contribution to Nature Communications.

Reviewer #2 (Remarks to the Author):

I thank the authors for the revised and improved version of the manuscript. I have a few further thoughts after considering the rebuttal and revised manuscript.

Response: Thank you again for your positive comments on our revised manuscript. We are grateful that the referee has spent time reviewing and is satisfied with most of our responses to the inquiries raised in the first referee report, and we appreciate these additional questions that do not really challenge or question the validity or technical aspects of our method, but help us to better present our method to the readers. We believe that we have addressed your concerns in the revised manuscript. The detailed point-by-point responses are given below.

(1) It is still unclear how the data is split and preprocessed. In the rebuttal, the authors state that the protein and drug embeddings are fed into the NFM along with the structural information features. What is still not clear and is crucial for there to be clarity on is are any links within the knowledge graphs between that particular protein and drug removed before the embeddings are removed? This could lead to a problem of trivial predictions where the embeddings between those two entities already encode that they should be linked.

Response: We thank the reviewer to point out the unclarity of the description for dataset processing. We have supplemented the following description in the evaluation protocols section of our revised manuscript.

Here, we process the whole knowledge graph into two parts: the task dataset and the supporting knowledge graph. In this work, the task dataset refers to the DTI dataset and the supporting knowledge graph refers to the drug-related information such as drug-drug interactions and protein-related information (e.g., protein-protein interactions) but without drug-target interactions. In the training process, (1) the DTI dataset is firstly split into the training set and the test set in each fold according to different scenarios (i.e., warm start, cold-start drugs, and cold-start proteins); (2) the supporting knowledge graph and DTIs in the training set are used to train the KGE model; (3) the embedding vectors deprived from the KGE model of the DTIs in the training set and the corresponding descriptors are used to train the NFM model. Finally, the prediction scores of the trained NFM model are used to evaluate the performance of our proposed framework. In

summary, the data used in the training process only includes drug-related information, protein-related information and DTIs in the training dataset.

In short, the potential mistake (due to inadvertent data processing) does not happen in our case.

(2) I would still like to see some hypotheses as to why the model does not perform better for the cold start on drugs given that the knowledge graphs should be able to capture information about them.

Response: We thank the referee for this question, and we agree it is nice to have a hypothesis that could help readers to better understand our algorithm.

Given the situation that the RF outperforms other baselines for the cold start on drugs, we delve into closer inspections on the subtle differences in the predictions by KGE_NFM and RF on the Yamanishi_08's dataset and BioKG via analyzing the precision-recall curve (Fig. R1). Under the scenario of the cold start for drugs, we observed that RF behaves better than KGE_NFM on the Yamanishi_08's dataset (Fig.R1 (a)) at the threshold with low recall rate (red box in the upper left corner). However, the gap of the precision between KGE_NFM and RF is noticeably reduced on BioKG. A possible explanation for this observation is that it is difficult for KGE_NFM to predict the relationships between unknown drugs and many proteins based on the limited size of the DTI dataset and drug-related knowledge graph (Table. R1). According to these data, we can also infer that the predictive performance of KGE_NFM toward DTIs in the scenario of the cold start for drugs could be improved with more DTIs data and sufficient supporting drug-related knowledge graph.

In addition, considering that RF based on the molecular fingerprints is more likely to capture the similarity correlations between the drugs in the test set and those in the training set, the differences of the predictive performance between KGE_NFM and RF on the Yamanishi_08's dataset could be related to the molecular similarities. For the Yamanishi_08's dataset, RF prefers to give higher prediction scores for molecules with relative high similarities compared with KGE_NFM (red box in Fig. R2 (a) and Fig. R2 (b)). While for BioKG, the difference of predictions between RF and KGE_NFM is not that significant. These findings suggest that the similarity-based feature is probably more efficient than knowledge-based feature when the size of DTIs and drug-related knowledge graph is limited. Nevertheless, with the increasing number of DTIs in the training set and drug-related knowledge graph, the advantage of similarity-based feature is no longer significant, and KGE_NFM is able to achieve satisfactory predictive performance under the scenario of the cold start for drugs.

Figure. R1. Evaluation performance comparison between KGE, KGE_NFM and RF in the scenario of cold start for drugs. (a) The precision-recall curve of the Yamanishi_08's dataset and (b) the precision-recall curve of BioKG.

Table R1. Statistics of DTIs and KG on the Yamanishi_08's dataset and BioKG

	Drugs in DTIs	Proteins in DTIs	DTIs	Drug-related KG	Protein-related KG
the Yamanishi_08's dataset	791	989	5127	9468	80984
BioKG	6214	3442	28033	1406066	637780

Figure. R2. The distribution of similarity and predictions of KGE_NFM and RF on the Yamanishi_08's dataset and BioKG. (a), (b) The distribution on the Yamanishi_08's dataset and (c), (d) the distribution on the BioKG. The predictions are the results of the positive pairs on the test set. The similarity is calculated via the Morgan fingerprints of 2048 bits between the drugs of positive pairs in the test set and all the drugs in the training set.

(3) Lastly, on rereading the article, it seems the authors provide limited justification as to why they chose DistMult over the myriad of other KGE models in the literature. How would the results change in TransE or RotatE was chosen instead?

Response: Thank you for your comment. Actually, the choice of DistMult is based on the observed superior predictive power of DisMult_NFM for the DTI datasets considered in this work. Here, we provide these additional results to substantiate the claim. Particularly, we provide the evaluated performance of the three KGE methods (TransE, RotatE, DistMult) and our

framework-enabled counterparts (i.e., TransE_NFM, RotatE_NFM and DistMult_NFM) on the Yamanishi_08's dataset and BioKG under the three sample scenarios (Figure R3, R4): warm start, cold-start drugs, and cold-start proteins.

As shown in Figures R3 and R4, DistMult_NFM proposed in the manuscript outperforms all the three KGE methods (TransE, RotatE, and DistMult) as well as the other two of our framework-enabled methods (TransE_NFM and RotatE_NFM) according to the metrics of AUROC and AUPR on the Yamanishi_08's dataset and BioKG under all the three sample scenarios. Moreover, TransE_NFM and RotatE_NFM gain significant boost in performance when compared with the bare TransE and RotatE in terms of AUROC and AUPR, which further validate the efficiency of our framework.

Figure. R3. Evaluation performance on the Yamanishi_08's dataset in three sample scenarios. Scenario of warm start: Fig.R3 (a) and (b); scenario of cold start for drugs: Fig.R3 (c) and (d); scenario of cold start for proteins: Fig.R3 (e) and (f). All the results were obtained by 10-fold cross-validations.

Figure. R4. Evaluation performance on the BioKG dataset in three sample scenarios. Scenario of warm start: Fig.R4 (a) and (b); scenario of cold start for drugs: Fig.R4 (c) and (d); scenario of cold start for proteins: Fig.R4 (e) and (f). All the results were obtained by 10-fold cross-validations.

Reviewers' Comments:

Reviewer #2:

Remarks to the Author:

The authors have addressed my comments in the revised version of the manuscript.